# Life before Stonehenge: The hunter-gatherer occupation and environment of Blick Mead revealed by sedaDNA, pollen and spores

**Samuel M. Hudson**[1]*, **Ben Pears**[1], **David Jacques**[2], **Thierry Fonville**[1], **Paul Hughes**[1], **Inger Alsos**[3], **Lisa Snape**[4], **Andreas Lang**[4], **Antony Brown**[1,3]

**1** Department of Geography and Environmental Science, University of Southampton, Southampton, United Kingdom, **2** School of Humanities, The University of Buckingham, Buckingham, United Kingdom, **3** Department of Natural Sciences, Tromsø University Museum, Arctic University of Tromsø, Tromsø, Norway, **4** Department of Geography and Geology, University of Salzburg, Salzburg, Austria

* S.M.Hudson@Soton.ac.uk

**Data Availability Statement:** All relevant data are within the manuscript and its Supporting Information files.

## Abstract

The Neolithic and Bronze Age construction and habitation of the Stonehenge Landscape has been extensively explored in previous research. However, little is known about the scale of pre-Neolithic activity and the extent to which the later monumental complex occupied an 'empty' landscape. There has been a long-running debate as to whether the monumental archaeology of Stonehenge was created in an uninhabited forested landscape or whether it was constructed in an already partly open area of pre-existing significance to late Mesolithic hunter-gatherers. This is of significance to a global discussion about the relationship between incoming farmers and indigenous hunter-gatherer societies that is highly relevant to both Old and New World archaeology. Here we present the results of plant sedaDNA, palynological and geoarchaeological analysis at the Late hunter-gatherer site complex of Blick Mead at the junction of the drylands of Salisbury Plain and the floodplain of the River Avon, on the edge of the Stonehenge World Heritage Site. The findings are placed within a chronological framework built on OSL, radiocarbon and relative archaeological dating. We show that Blick Mead existed in a clearing in deciduous woodland, exploited by aurochsen, deer and hunter-gatherers for approximately 4000 years. Given its rich archaeology and longevity this strongly supports the arguments of continuity between the Late Mesolithic hunter-gatherers activity and Neolithic monument builders, and more specifically that this was a partially open environment important to both groups. This study also demonstrates that sediments from low-energy floodplains can provide suitable samples for successful environmental assaying using sedaDNA, provided they are supported by secure dating and complementary environmental proxies.

## Introduction

The interaction of hunter-gatherers with farmers is a theme of central importance in global archaeology. In Britain the monumentalized landscapes of Neolithic farmers appear to present

**Funding:** The author(s) received no specific funding for this work. However, the corresponding author did receive funding from the University of Southampton for general fieldwork costs.

**Competing interests:** The authors have declared that no competing interests exist.

a fundamental discontinuity with hunter-gatherer landscapes, however, recently this has been questioned due in part to the short-lived signature of agriculture the early Neolithic record [1]. With this background we present here new environmental data on the major pre-Neolithic archaeological site located at the edge of the Stonehenge World Heritage Site (SWHS)—the Blick Mead site complex.

Blick Mead is situated on the northern edge of the River Avon floodplain, Wiltshire, surrounded by the famous Neolithic archaeological landscape that forms the SWHS (Fig 1). Excavations have revealed extensive Mesolithic archaeology including a high number of stone tools (>100,000), and a varied macrofaunal assemblage dominated by aurochsen [2, 3] adjacent to the floodplain, demonstrating on-site anthropogenic activity between 8000 and 3400 BCE (S1 Table). The site is significant as it has bearing on whether the Stonehenge monumental complex was established *de-novo*, within, or even in response to, an already inhabited and partially open landscape. It is important to understand the character and scale of pre-Neolithic activity

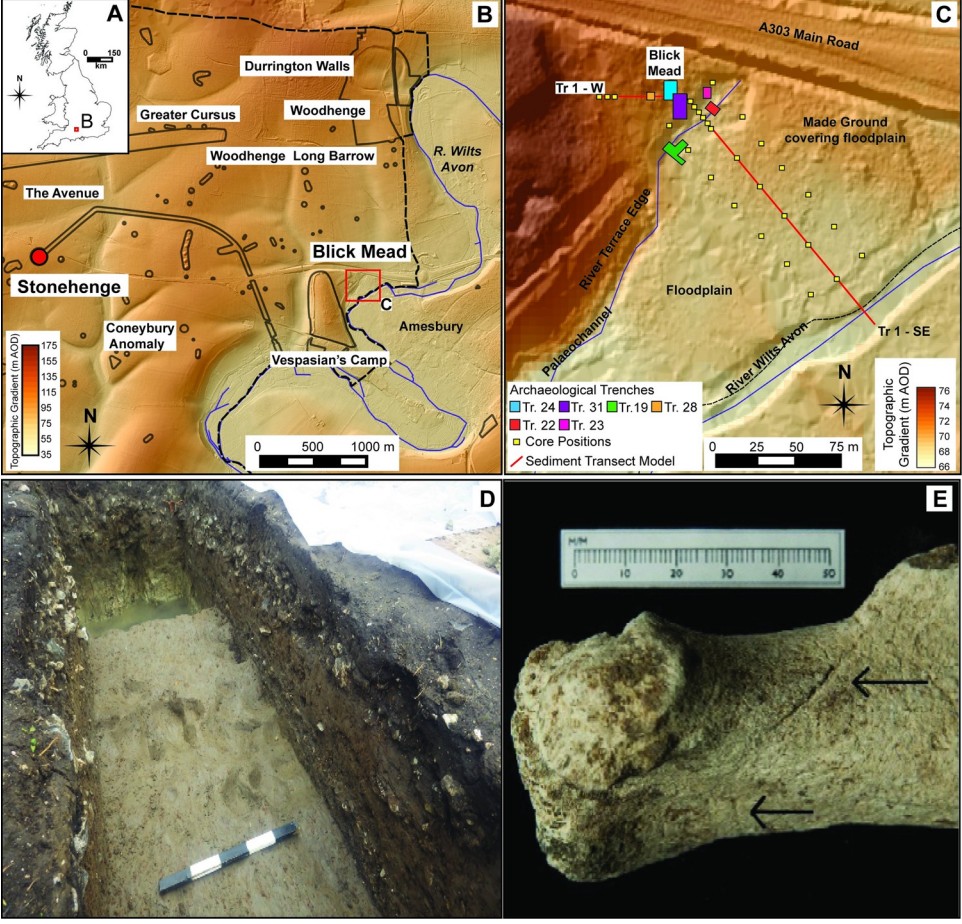

**Fig 1. Location, setting and key archaeology. A** UK location of the Stonehenge Environs. Map of UK derived from OS open source administrative boundary data © Crown copyright, (Boundary Line 2021). **B** Eastern section of the Stonehenge World Heritage Site with key archaeological sites. Basemap 1m Lidar DTM topographic gradient over Hillshade Model (All Lidar basemaps derived from open source UK environmental agency data © Crown copyright, Environmental Agency 2017). **C** Location of the Blick Mead Site on the edge of the Wiltshire Avon floodplain with archaeological trenches and positions of sediment cores and transect. Basemap 1m Lidar DTM topographic gradient (SU14SW; SU14SE) over Hillshade Model (Environment Agency 2017). **D** Position of in-situ auroch hoofprints within the Mesolithic alluvium (Photo D. Jacques). **E** Evidence of butchery cut marks on auroch faunal remains (Reprinted from [2] under a CC BY license, with permission from D. Jacques, original copyright 2018).

and the possible dynamic nature of the late Mesolithic environment. It has been hypothesized that the Mesolithic human populations based at Blick Mead and other sites in the region regarded this area as culturally important and both this and the partially open nature of the area lead eventually to its high status in the Neolithic and the construction of Europe's largest multi-monument symbolic landscape [3]. Such continuity has been contested [4] but has proven difficult to test due to the nature of sediments and poor preservation of palaeoenvironmental evidence within chalkland landscapes. Whilst closed-canopy forests contain many ecosystem benefits for hunter-gatherer communities, both ecologically and in terms of the utility, open areas must have also been important to attract large game, thereby facilitating hunting and increasing possible locations for ritual practice [5, 6]. The value of open grasslands reflects the shared nature of land-use at this time, but such areas were even more important to early farmers with a dominant pastoral-based economy [6].

The archaeological investigations at Blick Mead present a unique opportunity to significantly evaluate prehistoric landscape conditions using a multiproxy palaeoenvironmental approach. Firstly, it is still unclear as to the extent to which the wider landscape around Blick Mead and across the Salisbury Plain and the Stonehenge World Heritage Site was covered in closed-canopy forest in the Late Mesolithic. The traditional view proposed by British ecologist Sir Arthur Tansley, states that by the Late Mesolithic [mid-Holocene], the UK was covered by poly-climax forest, which was only rarely broken by deliberate forest burning and clearance [7]. Such forest clearance has been postulated by Innes *et al.* in the North Yorkshire Moors [8, 9], but so far there has been little direct evidence from Salisbury Plain. Alternatively, Sir Cyril Fox was one of the first to suggest that much of the prehistoric chalkland landscape in southern Britain may not have been heavily forested [10]. More recently, ecologists and palaeoecologists have developed this idea by suggesting that Late Mesolithic vegetation was more spatially variable and partially kept open by large ungulate grazing [6, 11–14]. More open areas could have been particularly common in chalkland environments, where shallow rooting depth of thin brown and rendzina soils increases the likelihood of naturally created forest clearances and grasslands from windthrow, beaver activity and lighting strikes [6, 15, 16]. Such clearances would have been opportunistically exploited by large ungulates for grazing and drinking, and therefore by Mesolithic people for occupation, hunting and gathering, which as a result subsequently suppressed the natural regeneration of woodland vegetation [14, 17].

The nature of shallow aerobic dryland chalk soils across much of southern Britain has resulted in the poor preservation of pollen and therefore a dearth of palaeoenvironmental evidence across large extents of these landscapes. Palynological reconstructions have however been achieved within organic deposits held within chalkland valleys [18–21], although interpretations have usually had clear geographical limitations. Additionally, chalkland areas that contain extensive deposits of *in-situ* aeolian (loess) have also been used to demonstrate that regional pollen diagrams lack the spatial resolution for the detection of isolated forest clearings [6, 22]. This makes the use of *sedaDNA* analysis potentially advantageous in these environments, particularly in areas such as Salisbury Plain and the Stonehenge World Heritage Site, where there is a distinct lack of suitable palaeo-vegetation records and traditional environmental proxies.

The limitations of pollen analysis have meant that previous research projects have used it alongside other palaeoenvironmental indicators. Dimbleby and Evans investigated several chalkland sites in the southern UK, including Durrington Walls, with molluscan evidence which suggested closed forest conditions by the Late Mesolithic [23], which was reinforced by further research at other chalkland locations [24–26]. Interpretations of the nature of the Late Mesolithic landscape have been identified across nearby chalkland floodplains. At Cranbourne Chase and the Upper Allen Valley (S1 Fig), pollen and snail analysis from the floodplain have

suggested patchier forest coverage than was previously envisaged [16, 27], with numerous open areas present by the Neolithic, supporting open grassland and pastoral exploitation [28]. French *et al.* came to similar conclusions from pollen and soil micromorphology analysis from floodplain sediments between Durrington Walls and West Amesbury [21]. These and other multi-proxy palaeoenvironmental studies have provided glimpses into the extent of localised open-grassland areas in the Stonehenge Landscape by the early Neolithic [29]. However, as Allen *et al.* has pointed out these studies have been highly skewed towards Neolithic-Bronze Age monuments and buried soils beneath them [30].

A second feature of the archaeological research is the idea that Blick Mead was a 'persistent place' [31] in the Mesolithic Stonehenge landscape [3]. It has been proposed that Blick Mead's past biodiversity, position at a river crossing point, and possibly ritual significance close to spring lines, led to the site becoming established as an area for repeated occupation. It is possible the presence of a unique assemblage of flora and fauna, alongside a source of mineral-rich spring water and a sheltered environment, meant that the site was 'well regarded' as a location for potential exploitation by Mesolithic communities [3]. Certainly, the presence of unusually high incidence of auroch bones in the macrofaunal assemblage, some with physical evidence of butchery, alongside numerous auroch hoofprints identified within Mesolithic soil horizons, suggests exploitation of this animal at Blick Mead [2, 3].

## Blick Mead site complex

The Blick Mead site is located on the northern edge of the floodplain of the Wiltshire Avon, around 150m from the main channel of the river, at the boundary between the alluvial floodplain, edge of an infilled palaeochannel with associated springhead and shallow valley side (Fig 1). In the local vicinity, the site is bordered by post-Mesolithic landscape development, including post-Bronze Age to medieval agricultural activity in the form of a marked agricultural lynchet to the north west, the extensive Iron Age Hill Fort known as Vespasian's Camp to the southwest [32], an artificial causeway from the 18th century landscaping of Amesbury Abbey to the south and extensive areas of made ground deposited in the 1960s during the construction of the present A303 road that passes to the north-east.

Archaeological excavations at Blick Mead began in 2005 and have continued annually to the present day, with numerous trenches extending across the low-lying floodplain, springhead and palaeochannel (Trench 19,24,31), and terrestrial hinterland (Trench 24,28). Within the fluvial-terrestrial margin significant Mesolithic archaeology has been identified including extensive artefactual evidence, faunal remains, auroch hoofprints all capped by a prehistoric stone surface. To the north west, on the terrestrial side the distinctive agricultural lynchet developed between the Middle Bronze Age and the later medieval period has buried, protected but compacted the earlier Mesolithic archaeology. Within these archaeological parameters, sedimentological sampling was conducted at a key interdigitation point between the fluvial, terrestrial and archaeological interfaces to extract *in-situ* sediment monoliths and sedaDNA samples. In addition, seventeen radiocarbon dates were also obtained including 12 from the wetland and five from the river terrace (S1 Table), alongside two additional OSL dates (S2 Table). In all these represent the longest Mesolithic dated sequence from any site in the UK. Both faunal and entomological analysis has been performed [2] and is included in the study for comparison with palynological and *seda*DNA analyses.

## Macroscopic palaeoenvironmental data from excavations

Of 2,430 bone fragments recovered, only 271 were identified in the zoological assessment [2] (S3 Table). The majority of bone fragments originated from Trench 19, within the sealed

Mesolithic context 59, with a large concentration in layer 77. Auroch (*Bos primegenius*) remains made up 57% of the assemblage and a very small number of domestic cattle bones were also found in the Neolithic contexts of Trench 24. Almost all auroch bones were represented and auroch was the only animal from Blick Mead showing any signs of butchery, with 5 bones exhibiting cut marks. Red deer (*Cervus elaphus)* was the second most common species found (17% of identified assemblage) along with smaller numbers of elk (Alces alces), roe deer (*Capreolus capreolus*), wild boar (*Sus scrofa*) and a single molar from a domestic dog (*Canis familiaris*) (S3 Table). Further faunal remains have been encountered in the Mesolithic deposits in Trenches 24 and 31 and are currently being examined by Peter Rowley-Conwy. Of the species identified so far, aurochsen, red deer and wild boar predominate in that order, reflecting the sequence in Trench 19.

An assessment of small vertebrates from the Mesolithic context 59 within Trench 19 was carried out in 2013 [2]. Summarised here, the 16 samples taken yielded a diverse small vertebrate fauna that includes fish, amphibians, lizards and small mammals (S4 Table). The majority of fragments originated from rodents, but many bulk samples also contained salmonid and pike bones, 80% of which showed signs of burning, likely representing meals discarded in or around a fire. The microfauna is dominated by woodland species such as bank vole (*Clethrionomys glareous*), yellow-necked mouse (*Apodemus flavicollis*) and marten (*Martes sp*.). However, there are some indications of grassland from the identification of *Microtus*, most likely field vole *Microtus agrestis*, and the presence of water vole which is thought to have occupied dry grassland until the Roman period when it was outcompeted by other grassland mammals [33].

From three bulk sediment samples taken from wetland trench 19 during excavation, a limited insect assemblage was identified (S5 Table) [2]. All samples originated from context 77, which has been dated to the Late Mesolithic, approximately 5200–4700 cal. BCE (S1 Table). Sample <5> was the richest, containing ten identified taxa. In this sample, the predaceous diving beetle [*Agabus*] and the water scavenger beetles *Helophorus brevipalpis* and *Laccobius minutus* were all suggestive of ponds, ditches or otherwise slow-moving water. Other species, such as *Tachyporus nitidulus* and many *Lathrobium* sp. are found in damp grass and moss tussocks. The seed weevil (*Protapion apricans*) feeds exclusively on the meadow plant red clover (*Trifolium pratense*). *Trifolium* was found in the *seda*DNA (see below). The leaf beetle (*Altica*) is also typical meadow taxa. Finally, the ant (*Myrmica schencki*) is an upland species of relatively dry sandy habitats in open areas and forests. Sample <6> yielded only three insect taxa: the water scavenger beetles *Enochrus* and *Octhebius minumus*, and the rove beetle (*Lathrobium*). Sample <7> yielded only the predaceous diving beetle. Taken together, the insects indicate the presence of a small, vegetation-choked pond or bog, with moss or grass tussocks on the margins, situated in a landscape that includes nearby herbaceous meadows.

## Results

### Stratigraphy, C14 and OSL chronology

In order to contextualize the localized stratigraphy identified within the archaeological excavations a 20m borehole survey was also conducted across the floodplain (Fig 2A and 2B). This confirmed the presence and extent of prehistoric horizons, through the identification of Mesolithic flint artefacts and these were deposited above weathered basal gravelly chalk deposits dated by OSL to the Late Pleistocene. In addition, the coring allowed the extent of a thin Mesolithic clay and palaeosol layer, and a cobbled flint surface overlain by further flinty loam alluvium and lynchet deposits to be extended beyond that confirmed during the archaeological excavations. The investigations also determined the extent of the former channel of the

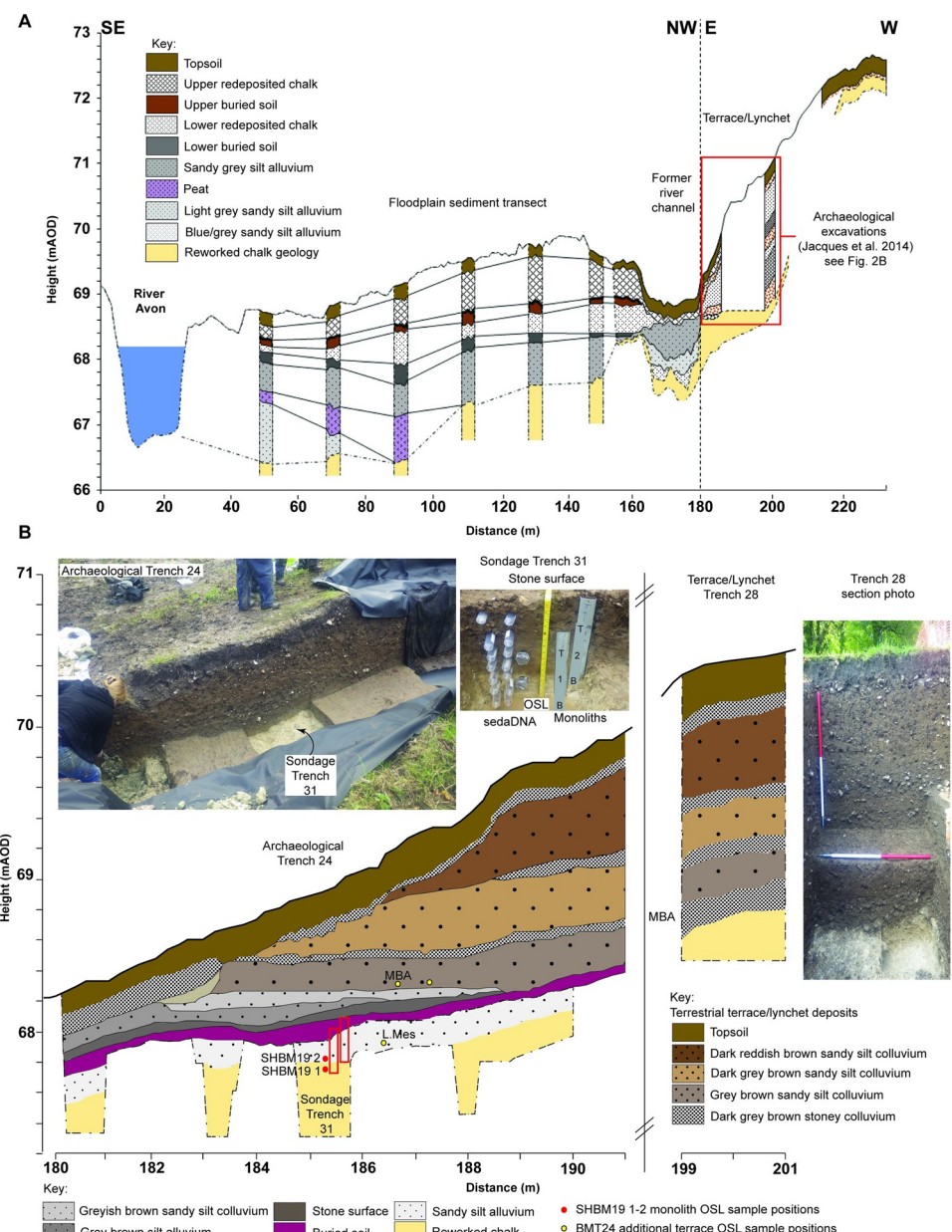

**Fig 2. A** Reconstructed sediment transect model across the floodplain of the Wilshire Avon between Blick Mead and the present river (see Fig 1). Transect shows extent of basal reworked Pleistocene gravelly chalk, fine grained alluvial deposits, peat deposits, extent of floodplain edge palaeochannel and buried soils associated with 18th and 19th century landscaping of Amesbury Abbey and construction of A303 road in the 1960s. **B** Detailed section drawing of fluvial-terrestrial interface between floodplain and Middle Bronze Age to late medieval lynchet. Section details the extent of basal reworked Pleistocene gravelly chalk, Mesolithic alluvium (containing auroch hoofprints), buried soil, stone surface as well as locations of sampling points in sondage 31, position of OSL and C14 dates (Photos S. Hudson).

Wiltshire Avon at the floodplain edge with distinctive sealed peat deposited in the Bronze Age [2], suggesting this is likely to have been an active channel during the Mesolithic. OSL sample locations can be seen in Fig 2B and the full suite of OSL and $C^{14}$ dates can be seen in S1 and S2 Tables.

The archaeological deposits from the *seda*DNA and pollen sampling sondage in trench 31 returned dates in stratigraphic order from the basal Late Pleistocene chalk deposits to the late Mesolithic period. The late Mesolithic dates for the middle of the sequence overlap with late Mesolithic $C^{14}$ dates from nearby trenches 19, 22 and 24. No evidence of the intrusion of later archaeological layers into the prehistoric horizons was detected. The dating is supported by the lithic evidence, which is wholly Mesolithic with the analysed stratigraphy extending beyond the sampling area into Trench 24 where a collection of small microliths and refits indicated a late 5th millennium date, suggesting no obvious contamination of DNA results from later human activity (Barry Bishop *pers com*).

In addition to the palaeoenvironmental information, extensive geoarchaeological evidence was also collected from the monoliths to determine depositional settings of the identified horizons. Combined Loss on Ignition (LOI) and elemental analysis (pXRF and ITRAX XRF) demonstrated that intra-horizon variability was limited but analytical variation could be seen between a fine-grained clay-silt alluvium (context 330) and overlying buried soil (context 328). The decrease in moisture content and sediment anoxia, reflected in changes in Mn, Fe and S between the alluvium and buried soil suggest a prolonged decrease in sediment saturation and fluvial input [34]. The basal reworked chalk is well represented by high Ca and carbonate levels ($LOI_{950}$) and the lack of higher fluctuations of Ca in the overlying layers indicated no reworking of the chalk in these horizons.

To further clarify depositional conditions within the analysed sequence, soil micromorphology was also performed on contexts 328, 330 and 334 (Fig 3). The micromorphological analysis demonstrates evidence of progressive low-energy accumulation within a fluvial-terrestrial interface. The basal reworked grey to dark grey chalk deposit (334) is distinctively carbonate clay rich with a dominance of calcite and silicates and a microstructure and fabric which suggests little to no *in-situ* disturbance in the sample area. This is supported by the lack of

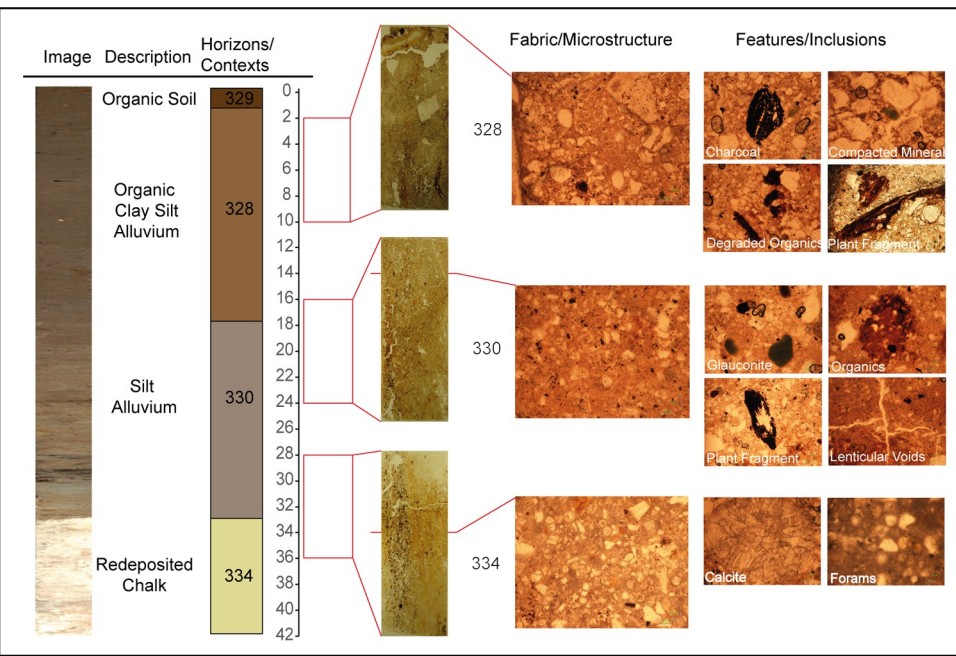

**Fig 3. Micromorphological analysis of sediment contexts 328, 330 and 334 in the analysed sequence.** Photos of each slide are shown alongside images of general fabric and microstructure characteristics alongside key organic and inorganic features and inclusions within the sediment. (All images are in plane polarised light at 10x magnification).

redeposited macro or micro-organic material from the overlying alluvial horizons. The sharp boundary between the basal chalk and the overlying alluvial deposits (330) and the progressive micromorphological characteristics also suggests prolonged stability in low-energy, undisturbed fluvial deposition at the floodplain edge. Texturally the horizon contained lower levels of calcite and chalk fragments and the increase in manganese and glauconite alongside the change in groundmass 'b' fabric and coarse to fine related distribution suggests undisturbed fine sedimentation and prolonged saturation. Unlike the chalk, this horizon demonstrated an increase in plant fragments and degraded organics, alongside greater micro amorphous black, brown and orange material associated with progressive *in-situ* degradation over time rather than by major horizon disturbance, although the increase in frequency of silt and clay void infills demonstrates the presence of a fluctuating groundwater conditions.

The more progressive change in textural characteristics at the top of the sequence suggests a gradual reduction in fluvial input and increased terrestrialisation, although fine sedimentation continues with occasional coarser quartz and flint deposition. The greater frequency of coarse and fine organics, particularly charcoal, plant macrofossils and trace lignified tissue, spores and turf indicate the presence of a remnant buried soil horizon which may have undergone a degree of compaction and truncation given the shift to a single spaced porphyritic and chitonic structure, but there is little evidence of the translocation of structure of inclusions from this horizon to the layers below. In addition, the reduction in silt and clay void infills, considerably lower iron nodules and increase in organics may have been the result of more prolonged phases of drier ground conditions. Overall, the combined geoarchaeological analysis suggested a well-established fluvial environment by the Mesolithic with progressive terrestrialisation of the floodplain from the Late Mesolithic onwards.

## Plant sedaDNA, pollen and spores

**SedaDNA.** From the samples taken from Trench 31, 2,392,544 raw sequence reads were obtained. After post-identification filtering, reads were trimmed to 1,477,389 and of the 41 remaining taxa, 11 were identified to species level, 20 to genus and 10 to family level (Fig 4). No aquatic taxa, ferns or mosses were recorded after post-identification filtering.

The redeposited chalk at the base of the sequence was dated to the Late Glacial/Early Holocene (9160 ± 1220BCE). From the bottom two samples- at 67.81 m and 67.77 m, very few DNA reads were obtained, only those from willow (Salicaceae). This was almost certainly due to very low template sedaDNA present in the weathered chalk, as high carbonate concentration is positive rather than negative for sedaDNA preservation [35]. The presence of willow suggests willow woodland around the waterbody and probably more widely on the floodplain.

The base of the alluvium overlying the chalk was dated to the Late Mesolithic (5690 ± 670BCE). The sample at 67.88 m revealed light woodland of apple (*Maleae*), dogwood (*Cornus)* and ivy (*Hedera helix)*, likely found on the river terrace wetland-dryland interface along with elm (*Ulmus*). *Willow* increases in both reads and replicates, likely inhabiting the floodplain with horsetail (*Equisetum*). The first graminoid taxa- *Agrostis/Alopecurus* (taxa cannot be separated due to sequence sharing) is found in small amounts.

The silt alluvium returned a Late Mesolithic date of 4650 ± 290 BCE and the sample at 67.93 m displayed a very slight decrease in tree/shrub taxa, but large increases in wetland and dryland forb taxa. Thistles (*Carduinae*), bindweeds (*Convolvulae*), dock/sorrel (*Rumex*) and the common stinging nettle (*Urtica dioica*) all appear in large numbers of PCR replicates, resembling a damp meadow forb community with areas of disturbed ground.

The samples at 68.03 m and 67.98 m show increased dry woodland with aspen (*Populus*) identified. Forb and graminoid diversity also increase, suggesting the continuance of an open

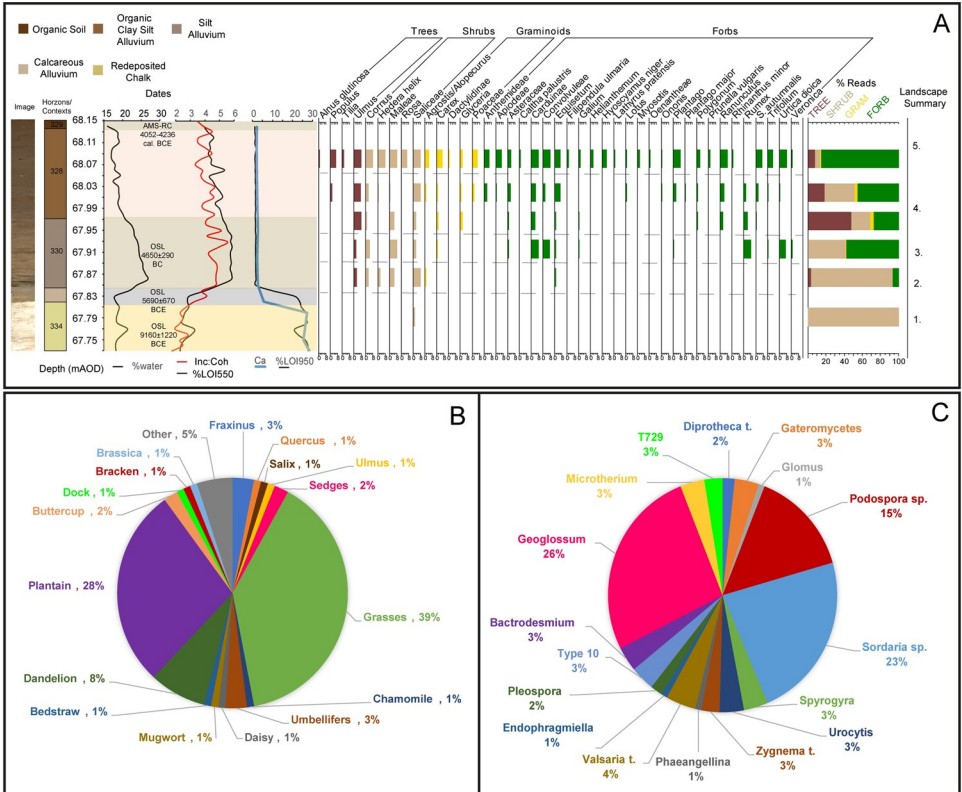

**Fig 4. The combined lithostratigraphic and environmental data. A** Selected lithostratigraphic data defining the main archaeological contexts and their associated dates. Shown alongside the full plant sedaDNA assemblage displayed as a histogram of the number of PCR replicates and taxa appeared in from 1–8 as well as a composite of total read percentages. The landscape summary determined from the plant sedaDNA evidence is as follows. 1- Willow woodland within floodplain. 2- More open local landscape with willow in floodplain alongside elm and increased shrubs on terrace edge. 3- Open local landscape with slight decrease in wood/shrub taxa, increases in wetland forbs and first appearance of sedge taxa. 4- Continued open local landscape with increased woodland and graminoid diversity, alongside increasing dryland forb community. 5- Continued open local landscape with clear increases in wet and dry woodland, graminoid and forb diversity. **B** Pollen assemblage taken from the Mesolithic layers of Trench 19 (see text). **C** Fungal and algal spore assemblage from the same samples.

local landscape. Further indications of soil disturbance are seen from the identification of plantain (*Plantago*) and continuing damp conditions are reflected in slight increases in horsetail and the appearance of taxa such as buttercups (*Ranunculus*), and sweet-grass (*Glyceria*)-a wetland grass of pond or river margins.

The organic upper alluvium was radiocarbon dated to 4236–4052 cal. BCE and the sample at 68.08 m showed large increases of all taxa types. A continued wetland presence in the form of willow, alder (*Alnus*), horsetail, sedges (*Carex*) and sweet-grass is suggested on the floodplain. Open conditions are indicated by an expanding community of wetland and dryland forbs, including clover (*Trifolium*), self-heal (*Prunella vulgaris*), rock-rose (*Helianthemum*), henbane (*Hyosycamus niger*), meadowsweet (*Filipendula ulmaria*), bedstraw (*Galium*), forget-me-knots (*Myosotis*) and yellow rattle (*Rhianthus minor*), with these forbs and others dominating read and replicate percentages, along with increases of grasses and sedges. Several of these plants have medicinal or food uses, the most interesting being henbane which has psychoactive properties and was used by early farmers in the Neolithic if not earlier [36]. We can also see an early reflection here of the typical chalk grassland and scrub (e.g. rock-rose and dogwood).

**Pollen and spores.** The sedaDNA results show a high degree of correspondence with pollen and spore analysis from the lower part of unit 2 from Trench 19 (monolith BM35 4b 80 cm) which correlates with contexts 330 and 333 in Trench 24 in which the auroch hoof marks were found. Of the 42 vascular plants identified in sedaDNA 21 are shared with pollen. Firstly, the pollen (65 types; S1 Dataset) is also dominated by grasses and plantains with willows, lime and elm at low levels. Secondly, the less common herbs and forbs are also shared by sedaDNA and pollen including daisy family (Asteraceae), bedstaw and docks/sorrel. Lastly, some relatively rare pollen types are also found in both datasets including ivy and clover. Overall, the interpretation of both assemblages is the same–a relatively large open grass-dominated clearing next to wet woodland and with mixed deciduous at some distance and probably upstream of the site. Both also reveal strong indicators of grazing and nutrient enrichment including clover, bedstraw, plantains and sheep's sorrel in both and nettles and forget-me-nots in the sedaDNA. It is also pertinent that several types found in the sedaDNA are typically poorly preserved as pollen (e.g. stinging nettles) or not distinguishable to the species level (e.g. *Maleae*). The fungal and algal spore results from the pollen core also show high levels of types that are typically associated with soils enriched with dung including *Sordaria* type and *Podospora* (Fig 4C) [37–39]. There is also evidence of standing water (*Spyrogyra*, *Zygnema*) and soil erosion (*Glomus*) [40], but also dry environments (e.g. T24) [41].

## The late hunter-gatherer environment

When combined the data suggest the development of damp meadow conditions, with an adjacent area of open grassland with nearby deciduous woodland. Forbs dominate sedaDNA read percentages and pollen counts, with woodland accounting for < 20% of reads at all depths but one. The dominant forbs in the *seda*DNA and pollen data- stinging nettle, plantain, dock/sorrel are all common on nutrient-enriched chalk grassland and indicate disturbed open and relatively dry ground [8, 11]. This is supported by the fungal and algal spore data from Trench 19, which is dominated by nutrient enriched grassland and/or coprophilous taxa but also has indications of wet shaded woodland and dry grassland soils. Between 67.82–67.98 m depth, the increases in floodplain activity suggested by the sedimentary analysis are likely to have led to a floodplain environment dominated by shorter-lived plants with high-stress tolerance and reproduction rates, so-called stress-tolerant-ruderals [42] such as the nettle, dock/sorrel and thistles which are seen in large numbers of PCR repeats at these depths. The more stable, drier environment seen above the silty alluvium allowed a variety of longer-lived meadow species to establish themselves and a more diverse meadow environment to develop. Taken together, from the evidence it could be suggested that an initial woodland-scrub dominated phase of vegetation made up of elm, willow, dogwood and apple is succeeded by a more stable damp meadow phase by the middle of the sequence. This could reflect the gradual creation of a clearing, either from natural causes such as river channel movement, or deliberate clearance by humans, which was then exploited for the large ungulate grazing of aurochsen.

The most obvious explanation for this long-lived open and eutrophic environment adjacent to the floodplain edge is the grazing of large ungulates such as auroch and deer, although other possibilities are discussed below. The dominance of aurochsen in the site faunal assemblage is unique in the UK and adjacent continental mainland. Other nearby Mesolithic sites in southern Britain show different faunal percentage patterns. Three Ways Wharf, Uxbridge in the Thames Valley is dominated by red deer [43], Faraday Road, in the Kennet Valley is dominated by wild boar [44] and Thatcham, Berkshire in the Thames Valley shows joint dominance of red deer and wild boar (see S1 Fig for locations) [45]. The high numbers of aurochsen may itself be an indication of environment, though now extinct, aurochsen are thought to have

grazed in lake/river-edge and had high daily water-requirement typical for cattle (*Bos*), damp grassland and similar wetland environments, which likely inhibited the expansion of woodland on these areas [46, 47].

## Discussion

The plant assemblage identified from sedaDNA and pollen analysis at Blick Mead is similar to the palynological analysis of other Mesolithic chalkland sites, in that a variety of shrubs and herbs, include those commonly found within chalk grassland. Comparative diverse herb assemblages can be in the South Downs [48], at Wimborne St Giles Profile 2 at Cranbourne Chase [16, 28] and to some extent that of Durrington Walls A15 [21]. What is significant is the lack of wider context tree/shrub taxa in comparison to these locations, which all show the presence of oak (*Quercus*) and hazel (*Corylus avellana*) and to a lesser degree birch (*Betula*) and pine (*Pinus*). The lack of oak, pine and birch in the *seda*DNA assemblage may be explained by Blick Mead's location <150 m from the present river channel, where due to the localised nature of *seda*DNA assemblages [49, 50] the plant taxa represented are likely to be only well representative of species growing <10 m from any sampling point. This means that floodplain adapted species such as willow and aspen are better represented. However, woodland taxa are also very limited in the palynological assessment, particularly birch and pine, confirming the true absence of these species from Blick Mead. Furthermore, the absence of hazel in both the sedaDNA and pollen counts and low reads and pollen counts of alder are also unusual, but may reflect willow outcompeting both on the floodplain locally.

There were four intrinsic measures of sedaDNA authenticity for this site. Firstly, since the sampled context was in groundwater contact with the underlying chalk, if sedaDNA had been translocated (leached) then it would be expected in the weathered chalk zone. However, this zone contained only willow sedaDNA which would be expected due to root penetration. Secondly, the sedaDNA derived plant assemblage does not contain tree taxa found on and around the site today, including beech (*Fagus*) and introductions including sycamore (*Acer*), fir (*Abies*) and cedar (*Cedrus*). Thirdly, there are taxa in the sedaDNA assemblage that are not recorded at the site today including lime, henbane, apple, bird's foot trefoil (*Lotus*) and yellow rattle. The targeted sediments were chosen for their sealed stratigraphic position, as indicated from the OSL chronology (S2 Table) which dated the formation of the overlying terrace deposits (cultivation lynchets) to the Middle-Late Bronze Age, protecting the Mesolithic horizons from later disturbance. Taxa appearing in negative controls were mostly common food and lab contaminants known from previous experience and can be seen in S6 Table. Finally, the micromorphological analysis suggested minimal bioturbation of the analysed sequence with no evidence of translocation between the horizons. The underlying alluvium was shown to be deposited in a prolonged, stable, low-energy fluvial depositional environment, whilst the upper buried soil was indicated to be somewhat compacted but otherwise undisturbed.

Overall, the plant assemblage at Blick Mead and other comparative chalkland and floodplain sites indicates partially open conditions and the persistence of elements of chalk grassland, possibly for extended periods during the Mesolithic. There is always the possibility of bias due to archaeological clearance activity of each site, but it seems rather more likely that heavily enclosed woodland was not present at these locations. Instead, chalkland floodplains like the Wiltshire Avon may have contained areas of marsh, meadow and forest clearings that were suitable for ungulate grazing, and part of wider mosaic woodland. This hypothesis is also supported by data from Late Mesolithic beetle percentages at alluvial sites which have reduced percentages of closed-canopy indicators and higher levels of dung beetles [13, 51].

Perhaps the most obvious explanation for the long-term persistence of an open clearing is due to grazing by large ungulates in areas close to their water and food supply. The clear evidence of human use of the site, including the stone surface and presence of cut aurochsen bones suggests that Blick Mead was also a persistent location for human ambush-hunting of aurochsen. Aurochsen are thought to have had similar watering needs to modern cattle [46], around 50 litres per animal, per day, and the availability of mineral-rich spring-water and favourable flora year-round may have meant that the species was more-or-less a constant presence on-site. Previous isotopic examinations of aurochsen teeth found at Blick Mead [2] did not indicate significant migration, and together with the hoofprints pressed into the archaeological sediment suggest a live population of aurochsen that were repeatedly exploited throughout the Mesolithic.

However, there are other not mutually exclusive possibilities, one being that hunter-gatherers kept the clearing open by removing trees. There is some evidence to support this, with 12 tranchet axes recovered from the site, along with large quantities of heavily burnt flint. However, the only direct evidence of deforestation is two tree throw pits dated to between 7960–7716 cal. BCE and 4336–4041 cal. BCE, which by themselves suggest either wind or possibly ungulates as a causal mechanism. That this site continued in use, however maintained, until at least 3400 BCE provides continuity with the first Neolithic activity in the area which was the construction of long barrows such as Amesbury 42 (c. 3520–3350 cal. BCE) [52] and the Greater cursus (c. 3630–3370 cal. BCE) [53]. It also means that it overlapped with the use of the nearby Coneybury Anomaly (3950–3780 cal. BCE)–a pit with a bone assemblage which it has been argued was created by a meeting between hunter-gatherers and early farmers [54]. This suggests that the first farmers into the area came into a partly open and occupied landscape and interacted with the local hunter-gatherer population.

A continuance of landscape use through the Late Mesolithic into the Neolithic is highlighted by Tilley, Edmonds and Evans *et al.* [55–57], who all note how Mesolithic forest clearings and pathways may prefigure later Neolithic routes and monuments constructed and managed by early farmers, in some circumstances with the enclosures mimicking the spatial pattern of clearings. A degree of landscape continuity supports theoretical frameworks surrounding the nature of early farming in Britain. Most notably, the 'garden cultivation system' [58] where it is argued that establishing new, large clearings in deciduous forest was unnecessarily difficult and not supported by the archaeobotanical evidence [59]. Instead, cattle would have been grazed in a series of long-established smaller clearings, in which natural growth of apple/pear and hazel was also encouraged for exploitation.

From the current evidence, it is proposed that Blick Mead represents the development of a floodplain edge clearing (Fig 5), with a pre-existing damp meadow environment that was then exploited for grazing by large ungulates, as indicated by taxa representing disturbed, nutrient enriched ground, such as *Urtica* and *Rumex*. Closer to the main river channel wet woodland would have been present with mixed deciduous woodland inhabiting the drier terrestrial margins. It seems likely from Blick Mead's location within one of the most archaeologically rich river valleys in the UK, that it is not an isolated environment in the Stonehenge landscape, but part of a series of more open areas along the chalkland floodplain. Such rich ungulate grazing may have been uncommon in Mid-to-Late Mesolithic Britain, and as such Blick Mead would have been returned to repeatedly for the exploitation of grazing animals such as aurochsen, red deer and wild boar. Certainly, the importance of the site regionally in the Mesolithic is attested to by its vast and diverse collection of Mesolithic struck flint, some of which, such as a slate microlith, may have originated from >100 km away [2] The site would have likely provided excellent access to a variety of habitats and floodplain resources, such as those of nearby riverine, woodland, and woodland edge environments. Moreover, the presence of nearby

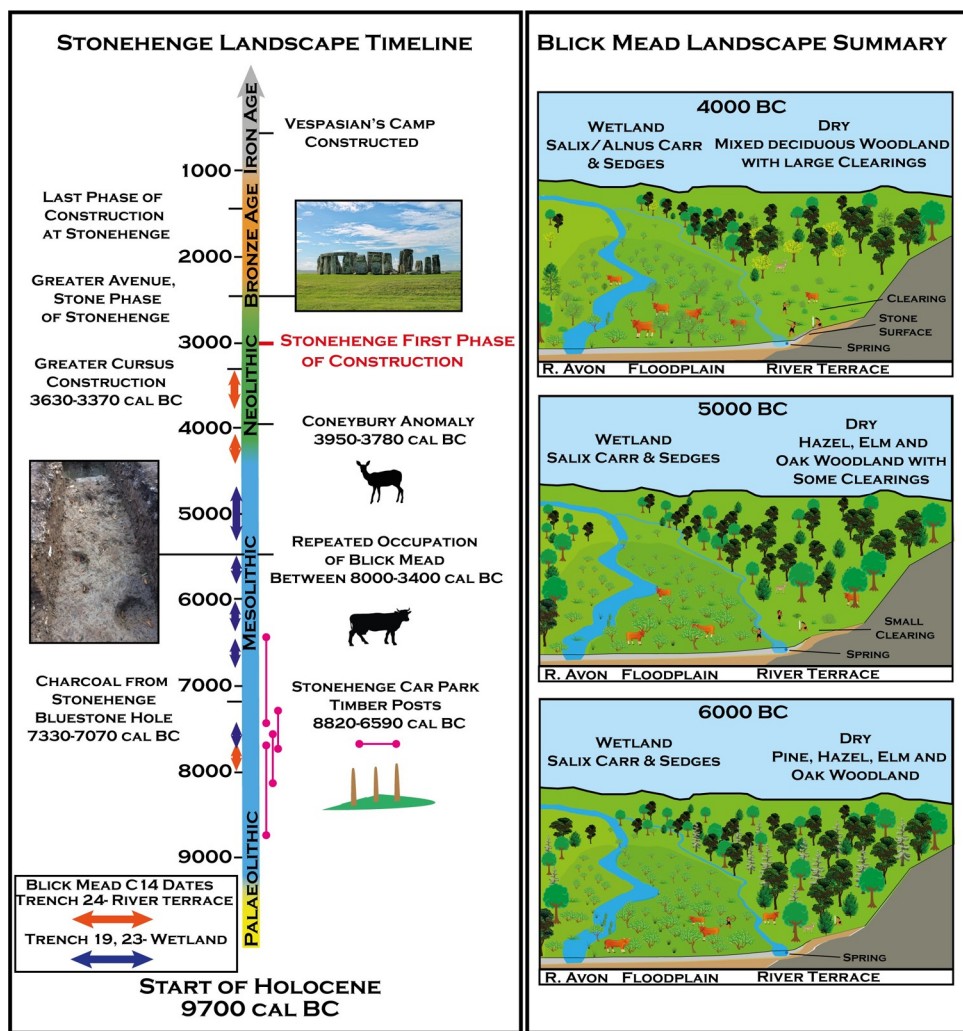

**Fig 5. A** Timeline of the Stonehenge landscape, including radiocarbon dates from Blick Mead and other significant SWHS archaeological sites. **B** A representation of the development of vegetation history at Blick Mead based on the palaeoenvironmental data.

spring-water, which aside from any functional use may have been used ritually to deposit flint and bone [3] as at other similar Mesolithic spring sites [60], was likely another attraction to the site. The creation of a stone surface in the Late Mesolithic (S1 Text) that covered both the wetland and terrace parts of the site may be linked to this spring activity, allowing easier access onto the wetland for hunting, butchery and object deposition. Furthermore, it is possible that the multiple channels adjacent to the surface were shallower than other parts of the river and divided by terrace islands, forming a crossing point for both animals and people. Together, these factors made Blick Mead a unique and favourable location with a wide subsistence landscape.

## Conclusion

From a landscape dominated by extensive land-use change including the development of Middle Bronze Age to late medieval agriculture and major upheaval during 18th to 19th-century landscaping and mid-20th century road building, we have been able to successfully extract

plant sedaDNA from waterlogged, sealed terrestrial Mesolithic contexts. The environmental data obtained from Blick Mead supports the theory that the wider Stonehenge landscape retained some persistently open areas before the extensive clearing seen in the Late Neolithic which were of great importance to late Mesolithic hunter-gatherers. These were particularly favoured locations (springs, wetland floodplains and cutoff channels adjacent to river terraces) and where ungulate grazing disrupted the natural succession and regrowth of woodland. Likely, these clearings were also locations for ritual activity which drew people in from wide areas. They in many ways were pre-adapted for small-scale animal pastures in the Early Neolithic. Whilst open woodland has also been indicated nearby at Durrington Walls and Stonehenge Down [21, 61], there remains a scarcity of Mesolithic environmental sequences in the Stonehenge World Heritage Site and so the results from Blick Mead are significant for understanding the dynamic nature of the pre-Neolithic landscape. Is it is also worth noting that almost the exact location in which Stonehenge was created had been the site of likely ritual activity marked by the presence of timber posts [62] during the earliest period of occupation by hunter-gatherers at Blick Mead. This all raises the possibility that the creation of Europe's largest monumental landscape not only had spatio-ecological continuity with the late hunter-gatherer landscape, but was in many ways was a development of it–as the maintenance of open areas by herbivore grazing and associated hunter-gatherer activity pre-adapted this, and some other geologically similar landscapes, to the pastoral economies of the early and later Neolithic.

## Materials and methods

### DNA, pollen and OSL sampling

Sampling in 2019 took place in newly opened trench 31 in the wetland area. The sampling procedure was as follows: The sediment face was cleaned with a trowel sterilised with bleach. Two 30 cm steel monoliths- Mon 1 (67.73–68.03 m) and Mon 2 (67.85–68.15 m) were inserted into the face and cut out with a knife. OSL tubes were inserted by hand, with the lower tube (OSL1) going in first, the ends of the tubes were immediately sealed with duct tape after going in and labelled. After sterilisation of the lid and lid thread with bleach, fourteen 50 ml falcon tubes for DNA were the last samplers to go in and the first to come out, to minimize the risk of contamination. After recording depths, the falcon tubes were removed and immediately stored at 4 degrees Celsius. OSL tubes were then removed and placed in light-proof bags, followed by the steel monoliths which were also stored at 4 degrees Celsius (Fig 2B, photo). No permits were required for the described study, which complied with all relevant regulations.

### DNA extraction, amplification and sequencing

DNA extraction followed protocols followed Alsos *et al*. [63] using the QIAGEN DNeasy Powersoil Kit but adapted for smaller input volume of < 0.35mg and was performed in a dedicated ancient DNA laboratory at the University of Southampton. PCR amplification used generic primers of the trnL P6 loop of the plant chloroplast genome and exact PCR procedure followed Alsos *et al*. [64]. Two PCR negative controls and one positive control were carried out. Eight individually tagged PCR repeats were made for each sample to increase the chance of detecting taxa represented by low quantities of DNA, as well as to increase confidence in the taxa identified. Paired-end sequencing was performed on an Illumina HiSeq 2500 platform using TruSeq SBS Kit v3 (FASTERIS SA, Switzerland). All next-generation sequence data were aligned, filtered and trimmed using the OBITools software package [65] using similar criteria as Alsos *et al*. [64]. The resulting barcodes were assigned to taxa using the *ecotag* program and four independent reference datasets. One reference contained arctic and boreal sequence [66],

one the NCBI nucleotide database (January 2021 release), one the PhyloAlps database [67] and finally the NorBOL database (http://www.norbol.org/). The resulting identifications were merged and filtered, retaining barcode sequences if they were identified to 100% in at least two reference sets and had at least 10 reads across the entire dataset. False positives relating to common PCR errors and food contaminants were removed based on 'blacklists' built up from previous research at Trømso Museum, as well as taxa identified above family level. For the last step of filtering, the frequency of PCR repeats in samples compared to negative controls was examined. Sequences were retained if they had an overall frequency of PCR repeats in samples at least twice as high as in that in negative controls. We present the data semi-quantitatively as the proportions of PCR repeats 1–8 and total reads. Where a reference dataset was unable to assign to specific species due to sequence sharing, we have listed all species that share the sequence.

## Pollen and spores

Pollen and spore processing initially followed standard procedures with the removal of carbonates, acetolysis, and hydrofluoric acid digestion. However, early attempts produced too little pollen to count. Subsequently a large volume was used (2 ml) and a process of elutriation (swirling) was added to concentrate pollen. Only one level produced enough pollen to count and a spectrum that was not affected by differential pollen destruction. It was therefore decided to count this level to an extended sum of over 1500 grains using x600 magnification and x1000 for determinations of difficult types. Standard keys were used but mostly comparison with reference material held both in the University of Southampton and also at Tromsø Museum, Norway. Spores were classified from the collected works of Van Geel and the collection held at the Hugo de Vries Laboratory University of Amsterdam and now available online through the NPP-ID database [68] and other comparative studies [39].

## OSL dating

To complement the existing radiocarbon and artefactual chronology of the site complex, samples taken from Trench 24 (lower part of the terrace) trench 28 (higher part of the terrace) and trench 31 (DNA sampling area) were subject to fine and coarse grain OSL dating at the University of Salzburg. Further methodological details are included in the S2 Text.

## Sediment geochemistry

The two 30 cm monoliths (Mon 1 and Mon 2) taken directly from the archaeological sediment were subject to geochemical examination using ITRAX, a high-resolution multi-function core scanner that enables both X-radiography and X-ray fluorescence (XRF) analysis on any sediment profile [69]. Additional manual pXRF analysis was used for general sedimentological characterization of the sequence, to identify elemental concentrations associated with specific depositional processes and to compliment and quantify elemental characteristics determined in the ITRAX XRF analysis. Sample scans were conducted with a Niton XL3t GOLDD+ portable XRF (pXRF) analyser in laboratory conditions. A total of 11 samples at 5cm resolution were freeze dried and analysed within 15ml plastic sample pots covered with 160μm Ultralene film within the mounted test stand. Readings were taken twice using the 'Soils' and 'Geochemistry' modes with a dwell time of 60 seconds composed of 15s 'main' filter (0-100keV), 15s 'low' filter (0-40keV), 10s 'high' filter (40-100keV) and 20s 'light' filter (0-10keV). Each setting was run at its factory set length to ensure quick scans. Results presented in this study are from the 'Geochemistry' setting and use pre-existing factory calibration settings. Loss-on-Ignition (LOI) and magnetic susceptibility analysis (MS) followed standard procedures [70].

## Soil micromorphology

Alongside sedimentological analysis of the Blick Mead sequence three micromorphological slides were produced by the University of Stirling (http://www.thin.stir.ac.uk/methods.html) from an undisturbed monolith sample within Trench 31, each measuring 80mm by 40mm. The monolith samples were initially acetone dried, impregnated with epoxy resin then cut and bonded to glass slides. These were then lapped until a thickness of 30μm and finally polished using 3μm diamond paste.

Each of the slides was analysed and described with a Nikon Eclipse 60i petrological microscope at a range of magnifications and light sources, in order to determine coarse rock and mineral components, coarse and fine organics, pedofeatures, microstructure, fabric arrangement, groundmass and relative soil distribution. Descriptions of the slides were based on [71–74] and important textural and contextual images were taken using a fixed microscope Nikon camera and saved in JPEG and TIFF formats. Key microstructural variation, features and inclusions identified are demonstrated in Fig 3. The summary soil descriptions are tabulated in S7 Table and extended results can be seen in the S3 Text.

## Supporting information

**S1 Text. Archaeological context of Blick Mead.**
(DOCX)

**S2 Text. OSL methodology.**
(DOCX)

**S3 Text. Extended description of micromorphological results.**
(DOCX)

**S1 Table. Radiocarbon dates.**
(DOCX)

**S2 Table. OSL dates.**
(DOCX)

**S3 Table. Large vertebrate analysis.**
(DOCX)

**S4 Table. Small vertebrate analysis.**
(DOCX)

**S5 Table. Insect analysis.**
(DOCX)

**S6 Table. DNA negative controls and filtered sequences.**
(DOCX)

**S7 Table. Micromorphological results.**
(DOCX)

**S1 Fig. Locations mentioned in text.**
(DOCX)

**S2 Fig. Geochemical and lithostratigraphic analysis.**
(DOCX)

**S3 Fig. Enlarged micrographs.**
(TIF)

**S1 Dataset. Pollen data.**
(XLSX)

**S2 Dataset. Plant DNA data.**
(XLSX)

**S3 Dataset. LOI, XRF and magnetic susceptibility data.**
(XLSX)

## Acknowledgments

We would like to thank all those involved who have helped us access and study the wonderful site of Blick Mead, including Mike Clarke, David Cornelius-Reid and family, Sir Edward and Lady Antrobus, The Piercey family, the Amesbury Community, the Amesbury Museum and Heritage Trust and Kathy Riley and the Polk County Archaeology Society. We also thank the British Ocean Sediment Core Research Facility (BOSCORF) for supporting the generation of the XRF core scanning. Extra thanks go to Rob Scaife for his attempts to analyse further pollen samples and to Pete Heintzman for his help with interpreting sedaDNA data. Finally, we would like to thank all referees for their useful comments and feedback.

## Author Contributions

**Conceptualization:** Antony Brown.

**Formal analysis:** Samuel M. Hudson.

**Funding acquisition:** Antony Brown.

**Investigation:** Samuel M. Hudson, Ben Pears, Thierry Fonville, Lisa Snape, Andreas Lang, Antony Brown.

**Methodology:** Inger Alsos, Antony Brown.

**Resources:** David Jacques, Inger Alsos, Antony Brown.

**Visualization:** Samuel M. Hudson, Ben Pears.

**Writing – original draft:** Samuel M. Hudson.

**Writing – review & editing:** Samuel M. Hudson, Ben Pears, David Jacques, Paul Hughes, Inger Alsos, Antony Brown.

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
