## [Decision Letter · Decision Letter 0]

10 Jan 2022

PONE-D-21-38931Life before Stonehenge: the hunter-gatherer occupation and environment of Blick Mead revealed by sedaDNA, pollen and sporesPLOS ONE

Dear Dr. Hudson,

Thank you for submitting your manuscript to PLOS ONE. After careful consideration, we feel that it has merit but does not fully meet PLOS ONE’s publication criteria as it currently stands. Therefore, we invite you to submit a revised version of the manuscript that addresses the points raised during the review process.

All comments must e addressed in detail before re-submission.

We look forward to receiving your revised manuscript.

Kind regards,

Peter F. Biehl, PhD

Academic Editor

PLOS ONE

Journal Requirements:

2. In your manuscript, please provide additional information regarding the specimens used in your study. Ensure that you have reported specimen numbers and complete repository information, including museum name and geographic location. 

For more information on PLOS ONE's requirements for paleontology and archaeology research, see https://journals.plos.org/plosone/s/submission-guidelines#loc-paleontology-and-archaeology-research.

4. We note that Figures 1,2 and 4 in your submission contain copyrighted images. All PLOS content is published under the Creative Commons Attribution License (CC BY 4.0), which means that the manuscript, images, and Supporting Information files will be freely available online, and any third party is permitted to access, download, copy, distribute, and use these materials in any way, even commercially, with proper attribution. For more information, see our copyright guidelines: http://journals.plos.org/plosone/s/licenses-and-copyright.

a. You may seek permission from the original copyright holder of Figures 1, 2 and 4 to publish the content specifically under the CC BY 4.0 license. 

Additional Editor Comments:

Your manuscript has now been seen by a referee, whose comments are appended below. You will see from these comments that while the referee finds your work of potential interest, the reviewer raised substantial concerns that must be addressed. In light of these comments, we cannot accept the manuscript for publication, but would be interested in considering a revised version that addresses these serious concerns.

We hope you will find the referees' comments useful as you decide how to proceed. Should presentation of further data and analysis allow you to address these criticisms, we would be happy to look at a substantially revised manuscript. However, please bear in mind that we will be reluctant to approach the referees again in the absence of major revisions.

Reviewers' comments:

Reviewer's Responses to Questions

**Comments to the Author**

1. Is the manuscript technically sound, and do the data support the conclusions?

Reviewer #1: No

2. Has the statistical analysis been performed appropriately and rigorously? 

Reviewer #1: N/A

3. Have the authors made all data underlying the findings in their manuscript fully available?

Reviewer #1: No

4. Is the manuscript presented in an intelligible fashion and written in standard English?

Reviewer #1: Yes

5. Review Comments to the Author

Reviewer #1: This is the latest in a series of reports of ongoing investigations of a Mesolithic site in the valley of the Wiltshire Avon in southern England. In many respects, Blick Mead is unremarkable. Though situated at the interface between chalkland and wetland, organic preservation is poor and there are no hearths or other convincing structural features (leaving aside tree-throw hollows and a putative flint ‘platform’). Yet the site has attracted an unusual amount of attention by virtue of its proximity to Stonehenge and claims by the investigators that Blick Mead was an aggregation site to which hunter-gatherers returned periodically over millennia to engage in feasting and other ritual activities, evidenced by large amounts of (unworked) burnt flint and the preponderance of aurochs (wild cattle) among the faunal remains – a ‘persistent place’ in the Mesolithic landscape of southern England and perhaps the beginning of the Stone Henge ritual complex. While many archaeologists might regard such ideas as speculative (even fantasy), they are nevertheless difficult to refute.

The Blick Mead excavations have targeted small areas around a former spring at the leading edge of a terrace and the adjacent river floodplain. While previous reports (Refs 2 & 3) dealt primarily with the archaeological evidence, the focus of the submitted manuscript is on the results of plant sedaDNA and palynological analysis of column samples from a profile through the sediments at the floodplain edge, supported by sedimentological, OSL and XRF analyses. The results are used to promote the view that the Mesolithic site occupied an open area or ‘clearing’ in deciduous woodland (or woodland-grassland mosaic) which provided opportunities for the exploitation of browsing and grazing ungulates, principally aurochs and red deer, evidenced by their presence among the faunal remains and the occurrence of hoofprints in palaeosurfaces.

For the most part, the manuscript is clearly written, and supported by informative colour illustrations and a range of supplementary information that specialists should find valuable. Given that the application of sedimentary ancient DNA (sedaDNA) analysis as a complement to more traditional, forms of palaeoecological investigation is still not mainstream in British (or indeed European) archaeology, this aspect of the paper can be considered innovative. However, the interpretations offered are not always (in my view) robust, and there are statements that need to be amplified and/or clarified to improve the presentation:

1. Line 125-128: “Certainly, the presence of unusually high prominence of auroch artefacts in the macrofaunal assemblage, some with physical evidence of butchery, alongside numerous auroch hoofprints identified within Mesolithic soil horizons, suggests a heightened exploitation of this particular animal at Blick Mead [Line 125-128]

COMMENT: Cut-marked bones resulting from butchery are not normally considered “artefacts”. Why not simply refer to them as “cut-marked bones”? It is difficult to see how hoofprints are necessarily linked to human exploitation, or the combination of a few cut-marked bones plus hoofprints (especially given the 3-4000 yr Mesolithic occupation history) is indicative of “heightened exploitation”.

2. Line 206-208: “The late Mesolithic [OSL] dates for the middle of the sequence align with late Mesolithic C14 dates from nearby trenches 19, 22 and 24”.

COMMENT: The 2-sigma errors on the OSL dates are very large (up to ±1280 yr). So, “align” would seem to be an overstatement.

3. Line 210-212: “… reinforcing the relative archaeological chronology provided by the stone tool assemblage and demonstrating evidence of no obvious, major contamination of DNA results from later human activity”.

COMMENT: What is meant by “relative archaeological chronology”. Was this established from stone artefact typology or stratigraphy or both, and when and by whom; moreover, how does this “demonstrate” the stratigraphic integrity of the sedaDNA results – give reference citation(s). I don’t have access to the 2018 monograph on Blick Mead (Ref. 2), but in his review of that monograph report Andrew David (whose opinion I value) states, “What is clear, but is perhaps not spelled out enough in the accompanying discussion, is that the Mesolithic deposit is a chronologically mixed accumulation” (David 2019: 440). This apparent contradiction needs to be addressed.

4. Line 331: “There were three intrinsic measures of sedaDNA authenticity for this site … [et seq.]”

COMMENT: The three reasons given for discounting vertical translocation of sedaDNA are observed differences between the (sedaDNA-derived) plant assemblages from the Mesolithic age (alluvial) sediments and the underlying weathered chalk (1) and the modern flora (2), and colluviation (linked to Bronze Age (BA) agriculture on the adjacent terrace) protecting the Mesolithic horizons from later disturbance (3). None of these is entirely convincing. BA and later colluviation might explain the absence of modern plant DNA but would not have protected the Mesolithic horizons from pedogenic and other ‘disturbances’ during the Late Mesolithic to MBA (c. 5700-1500 BCE). From the context/horizon descriptions in Fig. 3A, I suspect the alluvial sediments were strongly affected by pedogenesis over a prolonged period, and that more-or-less the entire profile (from context 329 downwards into 334) is a buried soil. To be blunt, stratigraphic interpretation of palynological and sedaDNA data from buried soils is arguably no more reliable than from a modern soil. Micromorphological (thin section) analysis is widely used to investigate the properties and processes in buried soils/sediments, and often essential for identifying the extent of translocation due to root penetration, percolation, and (perhaps relevant here) animal/human trampling. Was soil microscopy and micromorphology applied at Blick Mead? If so, the results should have been included in this report. If not, this is a serious omission, which needs to be rectified before publication of the sedaDNA and palynological data.

5. METHODS/SUPPORTING INFORMATION: Overall, the descriptions of materials and methods are adequate, except for sediment geochemistry and particularly XRF. The micro-XRF (iTRAX) system is described in reference 69 (which is acceptable), but more explanation should be provided for the handheld pXRF measurements. Why was it deemed necessary to use both pXRF and μXRF? Were the pXRF measurements taken in the field or in the lab and, if in the lab, how were the samples prepared? What settings were used with the Niton XL3t – calibration model, filters, spot size, measurement times, etc. What are the detection limits for different elements using those settings? Was an empirical calibration used based on CRMs, or just the values provided by the instrument without external calibration?

MINOR CRITICISMS (Grammar/Use of English/Word Choice/Terminology)

General – “auroch” or “aurochs” are (alternative) SINGULAR nouns for Bos primigenius; “aurochsen” is the PLURAL/collective noun;

Line 19 – missing <letter o="" stroke="" with=""> in Tromsø;

Line 37 – “drylands” is one word;

Line 40 – “open clearing” – the word “open” seems superfluous;

Line 59 – DELETE “over approximately 4000 years ago” [REDUNDANT];

Line 71 – DELETE “to particular areas” (REDUNDANT); “facilitating” would be better than “easing”;

Line 90 – “windthrow” is the technical term for the uprooting and overthrowing of trees by wind;

Line 98 – “… chalkland areas THAT contain …”;

Line 116 – DELETE “to” before “towards”, i.e. “… skewed towards Neolithic …”;

Line 119 – “… Blick Mead WAS a persistent place …” [past tense];

Line 122 – Missing “a” before “unique”;

Line 125 – “… high INCIDENCE of auroch …” (not “prominence”);

Line 128 – DELETE “particular” (REDUNDANT);

Line 146 – INSERT definite article (the) before “later medieval”;

Line 155 – “The zoological assessment … identified 271 bone fragments”. The meaning is unclear. This would be better written as, “Of 2430 bone fragments recovered, only 271 were identifiable to species (S3 Table) …”

Line 159-160 – “… auroch WAS the only animal …”;

Line 162 – “Red deer WAS the second …”;

Line 163 – Italicise “Sus scrofa”;

Line 171 – “salmonid and pike” are common nouns, therefore not capitalised;

Line 172 – Change “fauna” to “microfauna”;

Line 184 – “Lathrobium sp.” – “sp.” should not be italicised;

Line 203-204 – modify text to read, “… full suite of C14 and OSL dates can be seen …”;

Line 189-190 – “Taken together the insects INDICATE …” (present tense)

Line 207 – “Late Pleistocene”. Capitalise “Late”;

Line 211-212 – “… demonstrating evidence of no obvious, major contamination of DNA results …”. Shorten this to, “… demonstrating no obvious contamination of DNA results …”

Line 233-234 – misspelling of Salicaceae; Delete “only”, thus: “…due to very low template sedaDNA…”.

Line 285 – “… the data suggest …”. “data” is plural.

Line 289 – “nutrient-enriched” [hyphenated]

Line 292 – “Between the depths of 67.82-67.98 m …”. This would read better as, “Between 67.82-67.98 depth, …”.

Line 294 – “high-stress” and “so-called” [hyphenated]

Line 302 – DELETE “possible”.

Line 314 – “…inhibited THE expansion …”;

Line 346 – DELETE “of time”.

Line 351 – “closed-canopy” [hyphenated]

Line 358 – “mineral-rich” [hyphenated]; “favourable” [British spelling, for consistency];

Line 372 – REPLACE “is” with “it”;

Line 374 – CHANGE “…had to interact with …” TO “… interacted with …”;

Line 388 – “wet woodland” [not hyphenated];

Line 390-391 – INSERT “from” between “that” and “Blick”, and DELETE “that” at beginning of line 391.

Line 392 – DELETE “fairly” [REDUNDANT];

Line 397 – DELETE “as a whole” [REDUNDANT]

Line 406 – “favourable” [BRITISH spelling, for consistency];

Line 410 – “land-use” [adjectival, therefore hyphenated];

Line 411 – “19th-century” [hyphenated];

Line 418 – REPLACE “It is likely that” WITH “Likely,”;

Line 424 – redundant “that” [that that];

Line 434 – “THE sampling procedure …”;

Line 438 – “…SEALED with duct tape …”;

Line 440 – “… THE risk of contamination …”;

Line 455 – “THE resulting barcodes …”;

Line 471 – DELETE “also”;

Line 484 – Further methodological details are included in THE S2 Text. [n.b. missing definite article]</letter>

6. PLOS authors have the option to publish the peer review history of their article (what does this mean?). If published, this will include your full peer review and any attached files.

Reviewer #1: No

---

## [Author Response · Author response to Decision Letter 0]

21 Feb 2022

The response to the the request for corrections from the editor (below)

Journal Requirements:

All files should now be named correctly.

2. In your manuscript, please provide additional information regarding the specimens used in your study. Ensure that you have reported specimen numbers and complete repository information, including museum name and geographic location. 

For more information on PLOS ONE's requirements for paleontology and archaeology research, see https://journals.plos.org/plosone/s/submission-guidelines#loc-paleontology-and-archaeology-research.

A statement has been added into the manuscript

I cannot seem to edit the financial disclosure section, but here I state that I received no grants for this work. If I have previously indicated that I received funding for fieldwork for this work specifically then that is incorrect.

4. We note that Figures 1,2 and 4 in your submission contain copyrighted images. All PLOS content is published under the Creative Commons Attribution License (CC BY 4.0), which means that the manuscript, images, and Supporting Information files will be freely available online, and any third party is permitted to access, download, copy, distribute, and use these materials in any way, even commercially, with proper attribution. For more information, see our copyright guidelines: http://journals.plos.org/plosone/s/licenses-and-copyright.

a. You may seek permission from the original copyright holder of Figures 1, 2 and 4 to publish the content specifically under the CC BY 4.0 license. 

A copyright letter has been included. One image not referred to by the letter (of Stonehenge) has a copyright licence CC0.

The response to reviewer comments is as follows (attached response letter):

Responses to Referees Comments and Actions Taken (in bold)

1. Line 125-128: “Certainly, the presence of unusually high prominence of auroch artefacts in the macrofaunal assemblage, some with physical evidence of butchery, alongside numerous auroch hoofprints identified within Mesolithic soil horizons, suggests a heightened exploitation of this particular animal at Blick Mead [Line 125-128]

COMMENT: Cut-marked bones resulting from butchery are not normally considered “artefacts”. Why not simply refer to them as “cut-marked bones”? It is difficult to see how hoofprints are necessarily linked to human exploitation, or the combination of a few cut-marked bones plus hoofprints (especially given the 3-4000 yr Mesolithic occupation history) is indicative of “heightened exploitation”. Agreed. Actions taken: We have removed the term “artefacts” and the term ‘heightened’.

2. Line 206-208: “The late Mesolithic [OSL] dates for the middle of the sequence align with late Mesolithic C14 dates from nearby trenches 19, 22 and 24”.

COMMENT: The 2-sigma errors on the OSL dates are very large (up to ±1280 yr). So, “align” would seem to be an overstatement. Agreed, although the maximum error range for the Late Mesolithic contexts is ±670yrs at 2σ . Actions taken: have altered the sentence to “The late Mesolithic dates for the middle of the sequence overlap with late Mesolithic C14 dates from nearby trenches 19, 22 and 24”.

3. Line 210-212: “… reinforcing the relative archaeological chronology provided by the stone tool assemblage and demonstrating evidence of no obvious, major contamination of DNA results from later human activity”.

COMMENT: What is meant by “relative archaeological chronology”. Was this established from stone artefact typology or stratigraphy or both, and when and by whom; moreover, how does this “demonstrate” the stratigraphic integrity of the sedaDNA results – give reference citation(s). I don’t have access to the 2018 monograph on Blick Mead (Ref. 2), but in his review of that monograph report Andrew David (whose opinion I value) states, “What is clear, but is perhaps not spelled out enough in the accompanying discussion, is that the Mesolithic deposit is a chronologically mixed accumulation” (David 2019: 440). This apparent contradiction needs to be addressed. This view was based on research up to 2013. Since then further stratigraphic work has established a tighter stratigraphy in Trenches 24, 24c and 31 (where our DNA samples were located) from stone artefact typology, sediment stratigraphy. Actions taken: we have included further information regarding the stone artefact typology and chronological implications from Trench 24, referenced by personal communication with the lithics specialist from Blick Mead- Barry Bishop on the 26/01/22 

4. Line 331: “There were three intrinsic measures of sedaDNA authenticity for this site … [et seq.]”

COMMENT: The three reasons given for discounting vertical translocation of sedaDNA are observed differences between the (sedaDNA-derived) plant assemblages from the Mesolithic age (alluvial) sediments and the underlying weathered chalk (1) and the modern flora (2), and colluviation (linked to Bronze Age (BA) agriculture on the adjacent terrace) protecting the Mesolithic horizons from later disturbance (3). None of these is entirely convincing. BA and later colluviation might explain the absence of modern plant DNA but would not have protected the Mesolithic horizons from pedogenic and other ‘disturbances’ during the Late Mesolithic to MBA (c. 5700-1500 BCE). From the context/horizon descriptions in Fig. 3A, I suspect the alluvial sediments were strongly affected by pedogenesis over a prolonged period, and that more-or-less the entire profile (from context 329 downwards into 334) is a buried soil. To be blunt, stratigraphic interpretation of palynological and sedaDNA data from buried soils is arguably no more reliable than from a modern soil. Micromorphological (thin section) analysis is widely used to investigate the properties and processes in buried soils/sediments, and often essential for identifying the extent of translocation due to root penetration, percolation, and (perhaps relevant here) animal/human trampling. Was soil microscopy and micromorphology applied at Blick Mead? If so, the results should have been included in this report. If not, this is a serious omission, which needs to be rectified before publication of the sedaDNA and palynological data. The upper buried soil is we believe still stratified with minimal mixing due to waterlogging typical of hydromorphic soils and we have micromorphology to back this up. Actions taken: we have included a new paragraph and soil micromorphology figure (Fig 3) in the results along with a summary in the discussion and extended information in the S3 text and the S7 table. The soil micromorphology suggested sediment deposition in a low-energy, fluvial environment with no evidence of translocation between the horizons and minimal bioturbation. 

5. METHODS/SUPPORTING INFORMATION: Overall, the descriptions of materials and methods are adequate, except for sediment geochemistry and particularly XRF. The micro-XRF (iTRAX) system is described in reference 69 (which is acceptable), but more explanation should be provided for the handheld pXRF measurements. Why was it deemed necessary to use both pXRF and μXRF? Were the pXRF measurements taken in the field or in the lab and, if in the lab, how were the samples prepared? What settings were used with the Niton XL3t – calibration model, filters, spot size, measurement times, etc. What are the detection limits for different elements using those settings? Was an empirical calibration used based on CRMs, or just the values provided by the instrument without external calibration? Agreed. Actions taken: we have expanded the details on the pXRF methodology giving the detail requested and explained why we used both pXRF and μXRF.

MINOR COMMENTS

General – “auroch” or “aurochs” are (alternative) SINGULAR nouns for Bos primigenius; “aurochsen” is the PLURAL/collective noun; Agreed. Action taken: all plurals changed

Line 19 – missing in Tromsø; Agreed. Action taken: ‘o’ changed to ‘ø’

Line 37 – “drylands” is one word; Agreed. Action taken: all cases altered to one word

Line 40 – “open clearing” – the word “open” seems superfluous; Agreed. Action taken: Removed

Line 59 – DELETE “over approximately 4000 years ago” [REDUNDANT]; Agreed. Action taken: Removed

Line 71 – DELETE “to particular areas” (REDUNDANT); “facilitating” would be better than “easing”; Agreed. Action taken: “to particular areas” removed and “easing” changed to “facilitating”. 

Line 90 – “windthrow” is the technical term for the uprooting and overthrowing of trees by wind; Agreed. Action taken: changed “windblow” to “windthrow”.

Line 98 – “… chalkland areas THAT contain …”; Agreed. Action taken: added “that”.

Line 116 – DELETE “to” before “towards”, i.e. “… skewed towards Neolithic …”; Agreed. Action taken: “to” deleted.

Line 119 – “… Blick Mead WAS a persistent place …” [past tense]; Agreed. Action taken: added “was”.

Line 122 – Missing “a” before “unique”; Agreed. Action taken: added “a”.

Line 125 – “… high INCIDENCE of auroch …” (not “prominence”); Agreed. Action taken: “prominence” changed to “incidence”.

Line 128 – DELETE “particular” (REDUNDANT); Agreed. Action taken: “particular” deleted.

Line 146 – INSERT definite article (the) before “later medieval”; Agreed. Action taken: “the” added.

Line 155 – “The zoological assessment … identified 271 bone fragments”. The meaning is unclear. This would be better written as, “Of 2430 bone fragments recovered, only 271 were identifiable to species (S3 Table) …” Agreed. Action taken: rewritten as “Of 2430 bone fragments recovered, only 271 were identifiable to species in the zoological assessment [2] (S3 Table)”.

Line 159-160 – “… auroch WAS the only animal …”; Agreed. Action taken: added “was”.

Line 162 – “Red deer WAS the second …”; Agreed. Action taken: added “was”.

Line 163 – Italicise “Sus scrofa”; Agreed. Action taken: italicised.

Line 171 – “salmonid and pike” are common nouns, therefore not capitalised; Agreed. Action taken: capitalisation of both words removed.

Line 172 – Change “fauna” to “microfauna”; Agreed. Action taken: changed to “microfauna”.

Line 184 – “Lathrobium sp.” – “sp.” should not be italicised; Agreed. Action taken: “sp” no longer italicised.

Line 203-204 – modify text to read, “… full suite of C14 and OSL dates can be seen …”; Agreed. Action taken: Text changed as suggested.

Line 189-190 – “Taken together the insects INDICATE …” (present tense) Agreed. Action taken: “indicated” changed to “indicate”.

Line 207 – “Late Pleistocene”. Capitalise “Late”; Agreed. Action taken: Capitalised “late”.

Line 211-212 – “… demonstrating evidence of no obvious, major contamination of DNA results …”. Shorten this to, “… demonstrating no obvious contamination of DNA results …” Agreed. Action taken: Text changed as suggested.

Line 233-234 – misspelling of Salicaceae; Delete “only”, thus: “…due to very low template sedaDNA…”. Agreed. Action taken: Spelling error “Saliceae” corrected to “Salicaceae” and “only” deleted. 

Line 285 – “… the data suggest …”. “data” is plural. Agreed. Action taken: “suggests” changed to “suggest”. 

Line 289 – “nutrient-enriched” [hyphenated] Agreed. Action taken: hyphenated “nutrient enriched”.

Line 292 – “Between the depths of 67.82-67.98 m …”. This would read better as, “Between 67.82-67.98 depth, …”. Agreed. Action taken: Text changed as suggested.

Line 294 – “high-stress” and “so-called” [hyphenated] Agreed. Action taken: hyphenated “high stress” and “so called”.

Line 302 – DELETE “possible”. Agreed. Action taken: “possible” deleted.

Line 314 – “…inhibited THE expansion …”; Agreed. Action taken: “the” added.

Line 346 – DELETE “of time”. Agreed. Action taken: “of time” deleted.

Line 351 – “closed-canopy” [hyphenated] Agreed. Action taken: “closed canopy” hyphenated.

Line 358 – “mineral-rich” [hyphenated]; “favourable” [British spelling, for consistency]; Agreed. Action taken: “mineral rich” hyphenated and “favorable” changed to “favourable”.

Line 372 – REPLACE “is” with “it”; Agreed. Action taken: “is” changed to “it”.

Line 374 – CHANGE “…had to interact with …” TO “… interacted with …”; Agreed. Action taken: Text changed as suggested.

Line 388 – “wet woodland” [not hyphenated]; Agreed. Action taken: hyphen removed from “wet-woodland”.

Line 390-391 – INSERT “from” between “that” and “Blick”, and DELETE “that” at beginning of line 391. Agreed. Action taken: Text changed as suggested.

Line 392 – DELETE “fairly” [REDUNDANT]; Agreed. Action taken: “fairly” deleted.

Line 397 – DELETE “as a whole” [REDUNDANT]; Agreed. Action taken: ”as a whole” deleted.

Line 406 – “favourable” [BRITISH spelling, for consistency]; Agreed. Action taken: “favourable” changed to “favourable”.

Line 410 – “land-use” [adjectival, therefore hyphenated]; Agreed. Action taken: “land use” hyphenated.

Line 411 – “19th-century” [hyphenated]; Agreed. Action taken: “19th century” hyphenated.

Line 418 – REPLACE “It is likely that” WITH “Likely,”; Agreed. Action taken: Text changed as suggested.

Line 424 – redundant “that” [that that]; Agreed. Action taken: “that” removed.

Line 434 – “THE sampling procedure …”; Agreed. Action taken: “the” added.

Line 438 – “…SEALED with duct tape …”; Agreed. Action taken: “taped” changed to “sealed”.

Line 440 – “… THE risk of contamination …”; Agreed. Action taken: “the” added.

Line 455 – “THE resulting barcodes …”; Agreed. Action taken: “the” added.

Line 471 – DELETE “also”; Agreed. Action taken: “also” deleted.

Line 484 – Further methodological details are included in THE S2 Text. [n.b. missing definite article]; Agreed. Action taken: “the” added.

---

## [Decision Letter · Decision Letter 1]

11 Mar 2022

PONE-D-21-38931R1Life before Stonehenge: the hunter-gatherer occupation and environment of Blick Mead revealed by sedaDNA, pollen and sporesPLOS ONE

Dear Dr. Hudson,

Thank you for submitting your manuscript to PLOS ONE. After careful consideration, we feel that it has merit but does not fully meet PLOS ONE’s publication criteria as it currently stands. Therefore, we invite you to submit a revised version of the manuscript that addresses the points raised during the review process.

Please address the remaining comments in detail before re-submission.

We look forward to receiving your revised manuscript.

Kind regards,

Peter F. Biehl, PhD

Academic Editor

PLOS ONE

Journal Requirements:

Additional Editor Comments (if provided):

Please address the remaining comments in detail before re-submission.

Reviewers' comments:

Reviewer's Responses to Questions

**Comments to the Author**

1. If the authors have adequately addressed your comments raised in a previous round of review and you feel that this manuscript is now acceptable for publication, you may indicate that here to bypass the “Comments to the Author” section, enter your conflict of interest statement in the “Confidential to Editor” section, and submit your "Accept" recommendation.

Reviewer #1: (No Response)

2. Is the manuscript technically sound, and do the data support the conclusions?

Reviewer #1: Yes

3. Has the statistical analysis been performed appropriately and rigorously? 

Reviewer #1: Yes

4. Have the authors made all data underlying the findings in their manuscript fully available?

Reviewer #1: No

5. Is the manuscript presented in an intelligible fashion and written in standard English?

Reviewer #1: Yes

6. Review Comments to the Author

Reviewer #1: The authors have taken on board most of the criticisms raised in my original review. However, I am still not satisfied with the information provided on soil micromorphology. A summary table has been added (S7 Table). But there are no micrographs of the soil thin sections. These should be provided as supporting information to the verbal descriptions (lines 222-250).

7. PLOS authors have the option to publish the peer review history of their article (what does this mean?). If published, this will include your full peer review and any attached files.

Reviewer #1: No

---

## [Author Response · Author response to Decision Letter 1]

18 Mar 2022

Reviewer #1: The authors have taken on board most of the criticisms raised in my original review. However, I am still not satisfied with the information provided on soil micromorphology. A summary table has been added (S7 Table). But there are no micrographs of the soil thin sections. These should be provided as supporting information to the verbal descriptions (lines 222-250).

Actions Taken: Soil micrographs are included in the revised figure 3, apologies if these were not clear, we have added a label to indicate them. In addition, enlarged photos of the micrographs have been added as S3 Figure.

With regards to availability of the data, we have now added the raw data for the LOI, XRF and Magnetic Susceptibility as S3 Dataset.

---

## [Editor Report · Decision Letter 2]

28 Mar 2022

Life before Stonehenge: the hunter-gatherer occupation and environment of Blick Mead revealed by sedaDNA, pollen and spores

PONE-D-21-38931R2

Dear Dr. Hudson,

We’re pleased to inform you that your manuscript has been judged scientifically suitable for publication and will be formally accepted for publication once it meets all outstanding technical requirements.

Kind regards,

Peter F. Biehl, PhD

Academic Editor

PLOS ONE
---

## [Editor Report · Acceptance letter]

1 Apr 2022

PONE-D-21-38931R2 

Life before Stonehenge: the hunter-gatherer occupation and environment of Blick Mead revealed by sedaDNA, pollen and spores 

Dear Dr. Hudson:

I'm pleased to inform you that your manuscript has been deemed suitable for publication in PLOS ONE. Congratulations! Your manuscript is now with our production department. 

Kind regards, 

on behalf of

Dr. Peter F. Biehl 

Academic Editor

PLOS ONE